# Surgical Outcomes of Video-Assisted versus Open Pneumonectomy for Lung Cancer: A Real-World Study

**DOI:** 10.3390/cancers14225683

**Published:** 2022-11-19

**Authors:** Jizhuang Luo, Chunyu Ji, Alessio Campisi, Tangbing Chen, Walter Weder, Wentao Fang

**Affiliations:** 1Department of Thoracic Surgery, Shanghai Chest Hospital, School of Medicine, Shanghai Jiao Tong University, No. 241 West Huaihai Road, Shanghai 200032, China; 2Department of Thoracic Surgery, University and Hospital of Trust-Ospedale Borgo Trento, Piazzale Aristide Stefani 1, 37126 Verona, Italy; 3Department of Thoracic Surgery, University Hospital Zurich, CH-8091 Zurich, Switzerland

**Keywords:** pneumonectomy, thoracotomy, minimally invasive, conversion, survival

## Abstract

**Simple Summary:**

Video-assisted thoracoscopic surgery (VATS) is now a well-established approach for anatomical lobectomy in the treatment of early-stage lung cancer. However, it is more difficult and risky to do VATS for pneumonectomy, with a more than 20% conversion rate. The safety, feasibility and potential benefits of VATS pneumonectomy remain to be investigated. Therefore, we performed a real-world study to evaluate the safety and feasibility of VATS pneumonectomy for lung cancer patients more comprehensively, with special attention paid to conversion cases and appropriate patient selection. This is a retrospective study with a largest cohort from a single center. The conclusions can deepen our understanding of the risks and benefits of VATS pneumonectomy.

**Abstract:**

Background: The safety, feasibility and potential benefits of Video-assisted thoracoscopic surgery (VATS) pneumonectomy remain to be investigated. Methods: Patients receiving VATS or Open pneumonectomy during the study period were included to compare surgical outcomes. Propensity-score matched (PSM) analysis was performed to eliminate potential biases. Results: From 2013 to 2020, 583 consecutive patients receiving either VATS (105, 18%) or Open (478, 82%) pneumonectomy were included. Conversion from VATS to open was found in 20 patients (19.0%). The conversion patients had similar rates of major complications and perioperative mortality compared with the Open group. After PSM, 203 patients were included. No significant differences were observed in major complications and perioperative mortality between the two groups. For patients with stage pT2 tumors, the major complication rate in the VATS group was significantly lower than in the Open group (7.6% vs. 20.6%, *p* = 0.042). Compared with left pneumonectomy, the incidence of bronchopleural fistula (BPF) was significantly higher in right pneumonectomy for both VATS (0 vs. 16.7%, *p* = 0.005) and Open (0.7% vs. 6.5%, *p* = 0.002) approaches. Conclusions: Perioperative results of VATS pneumonectomy are non-inferior to those of the Open approach. Conversion to open surgery does not compromise perioperative outcomes. Patients with lower pT stage tumors who need pneumonectomy may benefit from VATS.

## 1. Introduction

Lung cancer remains the leading cause of cancer-related death worldwide [1]. Currently, anatomical lung resection (lobectomy and segmentectomy) is the mainstay treatment for early-stage non-small cell lung cancer (NSCLC) [2,3]. However, for patients with locally advanced diseases or centrally located tumors, pneumonectomy may still be necessary. Traditionally, pneumonectomy is mostly performed via open thoracotomy, as patients requiring pneumonectomy often have large or hilar masses, and thus, are at a higher risk of uncontrollable bleeding or difficult hilar dissection. Together with an extensive loss of lung parenchyma, pneumonectomy via open thoracotomy is associated with a higher incidence of complications compared to lesser resections [4]. Cardiopulmonary functions and patient quality of life can be significantly impaired, especially after right pneumonectomy [5]. With the advent of surgical technology and the accumulation of experience regarding minimally invasive surgery, attempts to use video-assisted thoracoscopic surgery (VATS) for pneumonectomy have begun, with the goal of reducing surgical trauma and improving surgical outcomes.

The first case of VATS pneumonectomy was accomplished almost 30 years ago [6]. However, to this day, VATS pneumonectomy is still not widely practiced because it is technically difficult and risky. Hennon and colleagues recently reported equivalent 5-year survivals after VATS and Open pneumonectomy [7]. However, the study was based on the National Cancer Database (NCBD), in which detailed perioperative results were lacking, except for overall mortality. Then, as a technically challenging procedure, VATS pneumonectomy has been associated with a high conversion rate. In a recent multi-center study, Yang et al. reported a 19% conversion rate in VATS pneumonectomy and a high 30-day perioperative mortality rate of 17% among patients who underwent conversion [8]. The outcomes in patients who experienced conversion are largely unknown. Besides, although in lobectomy, VATS is associated with better postoperative recovery and less complications than open surgery [9,10,11,12], evidence for the potential benefits of VATS in pneumonectomy is inadequate. It is still unclear whether factors such as tumor stage or laterality affect the safety and feasibility of VATS pneumonectomy. Thus, the aim of our study was to more comprehensively evaluate the safety and feasibility of VATS pneumonectomy for NSCLC patients performed at a high-volume center, with special attention being paid to conversion cases and appropriate patient selection.

## 2. Materials and Methods

### 2.1. Patient Selection

This was a real-world study in NSCLC patients having pneumonectomy at the Shanghai Chest Hospital. All consecutive patients who received open or VATS pneumonectomy for NSCLC during January 2013 and December 2020 were included. Patients who received completion pneumonectomy were excluded. The study was approved by the Shanghai Chest Hospital Institutional Review Board (IRB:IS22041).

During the preoperative assessment, all patients underwent a detailed interview, physical examination, chest CT scan, brain MRI, electrocardiography and pulmonary function tests. Positron emission tomography-computed tomography (PET-CT) and bronchoscopy, EBUS-TBNA or mediastinoscopy were recommended in case of suspected lymph node involvement in mediastinum.

The selection for VATS pneumonectomy was by individual surgeon’s decision on resectability and their experience in VATS at different times on each patient. VATS pneumonectomy was defined as anatomic resection for pulmonary malignancy without rib spreading or muscle severing. The access incision including a standard 3–4 ports approach, with a 3–5 cm working port in the 4th intercostal space, the camera port in the 7th intercostal space in the middle axillary line and the posterior port in the 8th intercostal space in the posterior axillary line. The resection was to be done with visualization on a video monitor instead of visualization directly through the incision, in which a complete resection of the tumor is intended. Open pneumonectomy was defined as anatomic resection for pulmonary tumor using posterolateral incision or muscle-sparing modified posterolateral incision. The resection was performed with direct visualization and no assistance from video instruments.

Tumor staging was based on the 8th edition of the AJCC cancer staging manual [13]. Tumor histology was re-classified according to the 2021 WHO classification of lung tumors [14].

### 2.2. Collection of Clinical Data

Detailed patient and tumor characteristics were retrieved from a prospectively kept database, including sex, age, pulmonary function, comorbidity, history of smoking, neoadjuvant therapies, tumor laterality and histology and pathologic TNM stage. Perioperative data including operation time and blood loss, number of dissected lymph nodes, resection status (R0-2), length of stay in intensive care unit (ICU), length of stay in hospital after surgery (LOS), major complications such as bleeding, pulmonary edema, acute respiratory distress syndrome, bronchopleural fistula (BPF), atrial fibrillation (AF), esophageal injury, gastrointestinal bleeding, 30-day and 90-day mortality. Survival status was obtained from clinical medical records or telephone follow-up.

In order to study the safety and feasibility of pneumonectomy accomplished via the VATS approach with different surgical difficulty, we further divided the patients into subgroups according to their tumor stage (pT2 and pT3-4) and compared perioperative results. Perioperative outcomes were also compared between right and left VATS pneumonectomy to determine if there was any difference. To study the impact of conversion on pneumonectomy patients, perioperative outcomes in VATS patients converted to open thoracotomy were then compared separately with the Open group. In order to minimize potential biases, a propensity-score matched (PSM) analysis was carried out to control confounding factors using the R Project Software (v. 2.14.1; https://www.r-project.org/mirrors.html, accessed on 1 October 2022). Patients who experienced conversion to Open were removed from VATS group, with the remaining patients in VATS group matched in 1:2 ratio with no replacement to patients in the Open pneumonectomy group. The propensity score of each patient was derived from a multivariable logistic model with covariates, including sex, age, neoadjuvant chemotherapy, comorbidity, laterality, pathologic T, N and TNM stage. The nearest-neighbor method was used with a caliper of 0.02.

### 2.3. Statistical Analysis

Categorical data were compared by the Chi-square test or the Fisher’s exact test when appropriate. Continuous variables were compared by unpaired Student t test or the Wilcoxon rank-sum test according to their distribution. Overall survival (OS) was estimated using the Kaplan–Meier method and compared by log-rank test between the Open and the VATS groups. Statistical significance was defined as *p* < 0.05. Statistical analyses were performed using the SPSS software (version 22; SAS Institute, Cary, NC, USA).

## 3. Results

Over an 8-year period, a total of 583 NSCLC patients who underwent pneumonectomy were included for the study. Among them, 478 patients (82.0%) underwent Open pneumonectomy. In the other 105 patients (18.0%), VATS was first attempted, with 20 (19.0%) patients converted to open thoracotomy during surgery. The baseline characteristics of the pre-matched patients are shown in Appendix A. The percentage of patients with stage I (15.2% vs. 7.9%, *p* = 0.020) was significantly higher in the VATS group than in the Open group. The percentage of patients who received right pneumonectomy via VATS was significantly higher than that via the open approach (26.7% vs. 16.1%, *p* = 0.016). The percentage of patients with pT2 tumors (60.0% vs. 51.9% *p* = 0.131) in the VATS group was also higher than in the Open group, but did not reach a statistical significance.

The proportion of VATS among all pneumonectomies increased from merely 10% to 40% during the study period, along with the increasing use of minimally invasive surgery for all lung cancer procedures. However, it was still much lower than the use of minimally invasive surgery for lesser resections (18.0% vs. 95.2%, *p* = 0.000) [15]. Although mean operation time was longer in the VATS group than in the Open group, it has gradually dropped from 266 min at the beginning to about 180 min at the end of the study period, while the mean operation time (140 min) for Open pneumonectomy remained stable throughout the entire study period (Figure 1).

Before PSM, there was no significantly difference in R0 resection (93.3% vs. 91.8%, *p* = 0.841), major complications (14.3% vs. 18.0%, *p* = 0.364), 30-day mortalities (2.9% vs. 1.9%, *p* = 0.460), or 90-day mortalities (3.8% vs. 2.3%, *p* = 0.326) between the VATS group and the Open group (Table 1) After PSM, a total of 203 patients were obtained for further analysis, including 71 (35.0%) in the VATS group and 132 (65.0%) in the Open group. There was no significant difference in baseline characteristics between these two matched groups (Appendix A).

Table 2 shows the perioperative outcomes of the patients after PSM. VATS was associated with significantly greater number of dissected lymph nodes than Open thoracotomy (18.6 ± 8.5 vs. 16.2 ± 6.4, *p* = 0.026). There was again no significant difference in R0 resection (94.4% vs. 91.7%, *p* = 0.675) and rates of complications (11.3% vs. 19.7%, *p* = 0.125), 30-day mortalities (2.8% vs. 3.8%, *p* = 1.000) or 90-day mortalities (2.8% vs. 4.5%, *p* = 0.822) between the VATS and the Open groups. Complete follow-up was achieved in 77.3% of the patients. At the end of follow-up, 97 (61.8%) patients died, with a median follow-up time of 31.3 months. There was no overall survival difference between the VATS and the Open groups (HR = 1.19, 95% CI 0.77–1.89, *p* = 0.418) (Figure 2).

During surgery, 20 patients (19.0%) were converted from VATS to Open thoracotomy. Perioperative outcomes of the converted patients are shown in Table 3. Conversion was associated with significantly prolonged operation time (171.6 ± 49.4 vs. 140.1 ± 46.1, *p* = 0.003) and increased blood loss (527.5 ± 784.3 vs. 220.4 ± 258.0, *p* = 0.097) compared with the Open group. But there was no significant difference in incidences of 30-day mortality (*p* = 0.873), 90-day mortality (*p* = 0.979) or postoperative major complications (*p* = 0.427) between the conversion and the Open group. Length of ICU stay (*p* = 0.966) and LOS (*p* = 0.778) were also similar between the two groups.

Table 4 shows the results of perioperative outcomes between VATS and Open pneumonectomy for pT2 and pT3-4 tumors. For patients with pT2 tumors, the overall complication rate in the VATS group was significantly lower than in the Open group (7.6% vs. 20.6%, *p* = 0.042). The length of stay in ICU (2.4 ± 1.9 vs. 3.3 ± 5.2, *p* = 0.190) and LOS (10.2 ± 4.9 vs. 12.4 ± 11.6, *p* = 0.189) in the VATS group was shorter than in the Open group, although the differences did not reach a statistical significance. For patients with pT3-4 tumors, although there was no difference in overall complication rate between the VATS and the Open group (18.7% vs. 15.2%, *p* = 0.606), the incidence of BPF in the VATS group was significantly higher than in the Open group (12.5% vs. 1.7%, *p* = 0.006). The 90-day mortality (9.4% vs. 1.3%, *p* = 0.026) and length of stay in ICU (3.8 ± 7.4 vs. 2.6 ± 2.1, *p* = 0.052) in the VATS group was also higher than the Open group.

Table 5 shows the results of perioperative outcomes between VATS and Open pneumonectomy for right-sided or left-sided patients. VATS left pneumonectomy was associated with a decreased incidence of major complications with a marginally significance compared with Open approach (8.2% vs. 16.7%, *p* = 0.088). Length of ICU stay (2.2 ± 1.9 vs. 3.0 ± 4.3, *p* = 0.168) and LOS (10.0 ± 4.8 vs. 11.6 ± 9.4, *p* = 0.203) in the VATS group were also a little bit shorter than in the Open group. However, VATS right pneumonectomy was related to a higher incidence of BPF compared with the Open approach (16.7% vs. 6.5%, *p* = 0.127), although without statistical difference. Additionally, compared with left pneumonectomy, the incidence of BPF was significantly higher in right pneumonectomies either via VATS (0 vs. 16.7%, *p* = 0.005) or Open (0.7% vs. 6.5%, *p* = 0.002) approaches (Appendix A).

## 4. Discussion

VATS is now the recommended surgical approach for resection of early-stage NSCLCs due to its perioperative benefits and similar oncological outcomes compared to open surgery [9,10,11]. In recent years, VATS has been attempted in more complex operations, including pneumonectomy. However, the frequency of VATS pneumonectomy remains limited because of concerns about its safety, technical feasibility and oncologic equivalence to open surgery. In this real-world study based on the largest single-center cohort ever reported, with the accumulation of experience regarding minimally invasive surgery, as many as 40% of pneumonectomies could now be accomplished by VATS, with the operation time gradually decreased to an acceptable range. Perioperative outcomes of VATS pneumonectomy were equivalent to open surgery. Conversion from VATS to open surgery did not compromise patient recovery. VATS pneumonectomy might be beneficial for patients with pT2 tumors. But right pneumonectomy was related to an increased incidence of BPF, both in VATS or open surgery.

Although the first case of VATS pneumonectomy was accomplished in the early 1990s [6], application of VATS in pneumonectomy remains limited. Even in Shanghai Chest Hospital, one of the largest thoracic surgery units in the world, the application of VATS in pneumonectomy was still much less frequent than in lesser resections [15]. However, with the accumulation of experience, that figure has increased from less than 10% before 2015 to nearly 40% at the end of the study. In this real-world study, VATS pneumonectomy patients were more likely to have pT2 tumors. Surgery for these patients is relatively easier than for those with higher pT stage tumors, allowing a higher chance for the use of minimally invasive approach and the opportunity to reduce postoperative complications and to enhance recovery. It is also interesting to notice that the percentage of right pneumonectomy among VATS patients was significantly higher than those among open surgery patients. Since right pneumonectomy is known to be associated with a higher risk of morbidity and mortality [16], this clearly reflected an attempt to reduce the surgical trauma with minimally invasive surgery in these patients.

Pneumonectomy carries the highest risk of morbidity and mortality among all routine pulmonary resections, making safety and feasibility the core issues for performing pneumonectomy under VATS [17,18]. According to the literature, pneumonectomy is associated with a significant incidence of perioperative mortality [7,8,19], with the 30-day and 90-day perioperative mortalities as high as 7% and 12% [20]. Recent studies reported no difference in 30- (7% vs. 8%, *p* = 0.75) or 90-day mortality (15% vs. 14%, *p* = 0.66) rates for minimally invasive surgeries compared to open approaches for pneumonectomy [7,8]. Consistent with previous studies, we also found that VATS pneumonectomy had similar 30-day mortality (2.9% vs. 1.9%, *p* = 0.460) and 90-day mortality (3.8% vs. 2.3%, *p* = 0.326) compared with open surgery. Additionally, our results from this real-world study showed a much lower mortality rate than previously reported, probably due to more experience accumulated in minimally invasive surgery at a high-volume center. In terms of feasibility for VATS pneumonectomy, use of the VATS approach increased from less than 10% to nearly 40% during the study period. Although the average operation time in the VATS group was significantly longer than in the Open group, it has decreased year by year, dropping from 266 min at the beginning to about 180 min in the end. Besides, there were no differences in estimated blood loss, major complications, length of stay in ICU or LOS between the VATS and the Open approaches. Our results indicated that VATS may be increasingly applicable in pneumonectomy and could achieve comparable perioperative results to open surgery. Thus, with the accumulation of surgical experience, minimally approach could become a safe and feasible choice for pneumonectomy in selected patients.

Even so, VATS pneumonectomy is still a challenging surgical procedure. In the recently released VIOLET trial, conversion rate was only 6.1% in VATS lobectomy for early-stage lung cancers [9]. However, the reported conversion rates in VATS pneumonectomy were unanimously much higher than those in VATS lobectomy, indicating that it is indeed technically demanding [21]. Battoo et al., reported a 20.3% conversion rate in their VATS pneumonectomy patients [19]. Hennon et al., found a conversion rate of more than 30% in the NCDB [7], but the 30-day and 90-day mortality rates were similar in patients undergoing minimally invasive or open surgery. Looking at our VATS group, 20 patients were converted to Open thoracotomy during surgery. As for the reasons of conversion, among the 20 conversions, 13 (65%) were due to difficulty in hilar dissection (dense adhesion; tumor invasion; bulky or calcified lymph nodes), and 7 (35%) were due to vascular causes (pulmonary artery, pulmonary vein or other vessel injuries). Although conversion was associated with significantly prolonged operation time and a marginal increase in blood loss comparing to the Open group, there was no difference in mortality, major complication rate or LOS. Our results is in accordance with the NCDB study, indicating that timely conversion in more difficult cases would not compromise surgical outcomes of the patients. With appropriate patient selection, it does little harm to start minimally invasively, even for challenging cases like pneumonectomy.

Even though previous studies indicated that the perioperative results for VATS pneumonectomy were not inferior to open procedure, it has never been investigated which subgroups of patients might benefit from VATS. In this real-world study with further subgroup analysis, VATS in pT2 stage pneumonectomy patient was found to be associated with a significantly lower overall complication rate (7.6% vs. 20.6%, *p* = 0.042) and a trend toward faster recovery compared with the Open approach. The conversion rate among pT2 stage patient was also lower than pT3-4 stage patient (7.6% vs. 20.6%). For pT3-4 stage patient, the 90-days mortality in the VATS group was higher than the Open group (9.4% vs. 1.3%, *p* = 0.026), together with a significantly higher BPF rate in the VATS group (12.5% vs. 1.7%, *p* = 0.006). Our results indicated that for lower pT stage cases, the VATS approach may be more feasible and easier to accomplish in pT2 stage tumors, and may be beneficial by reducing surgical traumas and thus postoperative morbidities in these patients. However, the safety of VATS pneumonectomy for pT3-4 stage patients still needs more data to validate.

Our results indicated that VATS for left pneumonectomy was a safe and feasible approach. It was associated with a decreased overall complication (8.2% vs. 16.7%, *p* = 0.088) and a trend toward faster recovery for shorter day in ICU and LOS, showing its potential benefit over open approach. On the contrary, right pneumonectomy is known to be associated with more postoperative complications [15,22]. But there have not been any studies on whether this increased risk could be reduced by the use of minimally invasive approach. In our study, we found that the incidence of BPF was significantly higher in right- pneumonectomy for both the VATS (0 vs. 16.7%, *p* = 0.005) and Open (0.7% vs. 6.5%, *p* = 0.002) approaches. The VATS approach not only failed to reduce the risk of BPF after right pneumonectomy, but was associated with an even higher BPF rate comparing to open surgery. A higher rate of BPF for the right side may be related to anatomy, potential devascularization, and technical factors. Compared with open surgery, which uses sharp dissection by scissors, VATS surgery relies more on energy instruments, resulting in a greater degree of devascularization to the bronchus. Additionally, compared to the left side, the right bronchial stump is less likely to be supported by mediastinal tissue. Compared to the VATS approach, it is easier to do prophylactic coverage of the bronchial stump with viable tissues, such as a pedicled flap, with open surgery than with VATS. Unfortunately, we do not have data to support these hypotheses. The intention of VATS was to reduce surgical trauma and make right pneumonectomy acceptable. However, the results in real-world and subgroup analysis did not support that presumption, which is an important finding of the study. Our results clearly indicated that the VATS approach should be selected very cautiously for pneumonectomy on the right side.

The present study has several limitations to be acknowledged. Firstly, it is a retrospective study, with unavoidable intrinsic biases. Secondly, perioperative outcomes may be influenced by surgeons’ techniques, experiences and preferences. In fact, these surgeries are usually performed by surgeons with extensive experience in minimally invasive techniques. Thirdly, we did not measure the potential benefits of VATS surgery, such as reduced pain score or QOL, because these data were not routinely collected unless patients were enrolled in prospective clinical trials.

## 5. Conclusions

In conclusion, VATS for pneumonectomy is a still a challenging procedure with a long operation time and high conversion rate. However, with the advent of surgical techniques, it could become a safe and feasible approach for pneumonectomy in experienced hands. Especially for patients with early-stage diseases on the left side, VATS is more likely to be beneficial if a pneumonectomy is necessary. Still, it does not help reduce the risks associated with higher stage tumors or right pneumonectomy.

## Figures and Tables

**Figure 1 cancers-14-05683-f001:**
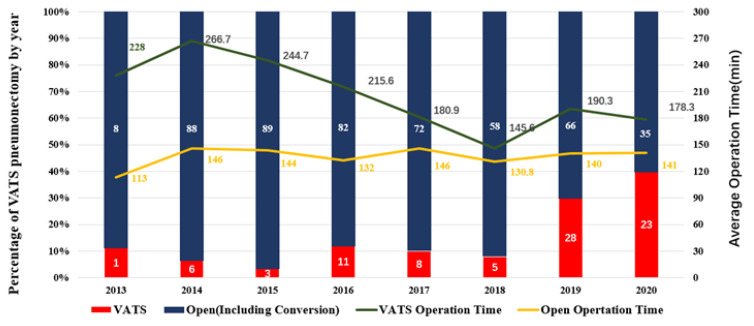
Changes in percentage of VATS pneumonectomy and average operation time during 2013–2020. The height of the red column represents the percentage of VATS pneumonectomy. The yellow curve represents the average time of Open pneumonectomy, while the green curve represents the mean time of VATS pneumonectomy.

**Figure 2 cancers-14-05683-f002:**
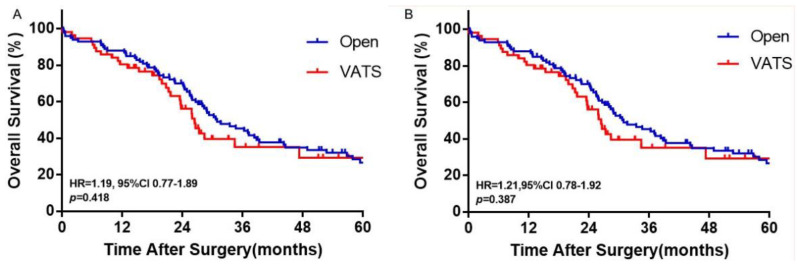
Overall survivals shown in Open and VATS pneumonectomy patients. (**A**) Overall survivals shown by Kaplan—Meier curves in Open and VATS pneumonectomy patients. (**B**) Cancer—specific survivals shown by Kaplan–Meier curves in Open and VATS pneumonectomy patients.

**Table 1 cancers-14-05683-t001:** Perioperative outcomes between Open and VATS pneumonectomy group patients before propensity-score matching.

Characteristic	Open n = 478	VATS n = 105	*p*
Operative time, min	140.1 ± 46.1	190.2 ± 67.3	<0.001
Blood loss, ml	220.4 ± 258.0	282.4 ± 410.0	0.140
Number of dissected LN	17.0 ± 7.1	17.7 ± 8.1	0.362
R0 Surgery, %	439 (91.8)	98 (93.3)	0.841
Major complications, %	86 (18.0)	15 (14.3)	0.364
Atrial Fibrillation	42 (8.8)	4 (3.8)	
Bleeding	11 (2.3)	1 (1.0)	
ARDS	10 (2.1)	0	
Infection	9 (2.5)	3 (2.9)	
Bronchopleural Fistula	8 (1.7)	7 (6.7)	
Chylothorax	3 (0.6)	0	
Esophageal injury	1 (0.2)	0	
Gastrointestinal bleeding	2 (0.4)	0	
Days in ICU	3.0 ± 4.1	2.9 ± 4.7	0.917
LOS, day	11.7 ± 9.3	10.5 ± 5.7	0.177
30 days mortality, %	9 (1.9)	3 (2.9)	0.460
90 days mortality, %	11 (2.3)	4 (3.8)	0.326

Abbreviations: LN = lymph node, LOS = length of stay. Categorical data are expressed as number (%) and continuous data as mean ± SD or median (interquartile range).

**Table 2 cancers-14-05683-t002:** Perioperative outcomes between Open and VATS pneumonectomy patients after propensity-score matching.

Characteristic	Open n = 132	VATS n = 71	*p*
Operative time, min	137.4 ± 46.8	188.2 ± 67.1	<0.001
Blood loss, ml	222.1 ± 231.4	200.0 ± 145.9	0.507
Number of dissected LN	16.2 ±6.4	18.6 ± 8.5	0.026
R0 Surgery, %	121 (91.7)	67 (94.4)	0.675
Major complications, %	26 (19.7)	8 (11.3)	0.125
Bronchopleural Fistula	5 (3.8)	3 (4.2)	
Atrial Fibrillation	10 (7.6)	3 (4.2)	
Bleeding	1 (0.8)	1 (1.4)	
ARDS	4 (3.0)	0	
Infection	3 (2.3)	1 (1.4)	
Esophageal injury	1 (0.8)	0	
gastrointestinal bleeding	2 (1.5)	0	
Days in ICU	3.3 ± 3.8	2.6 ± 3.5	0.214
LOS, day	12.9 ± 14.6	10.2 ± 5.3	0.136
30 days mortality, %	5 (3.8)	2 (2.8)	1.000
90 days mortality, %	6 (4.5)	2 (2.8)	0.822

Abbreviations: LN = lymph node, ARDS = Acute Respiratory Distress Syndrome, LOS length of stay, Categorical data are expressed as number (%) and continuous data as mean ± SD or median (interquartile range).

**Table 3 cancers-14-05683-t003:** Perioperative outcomes between Open pneumonectomy and VATS conversion to open patients.

Characteristic	Open n = 478 (%)	Conversion to Open n = 20 (%)	*p*
Operative time, min	140.1 ± 46.1	171.6 ± 49.4	0.003
Blood loss, mL	220.4 ± 258.0	527.5 ± 784.3	0.097
Number of dissected LN	17.0 ± 7.1	15.9 ± 6.7	0.512
R0 Surgery, %	439 (91.8)	19 (95.0)	0.929
Major Complications, %	86 (18.0)	5 (25.0)	0.427
Days in ICU	3.0 ± 4.1	3.0 ± 4.4	0.966
LOS, day	11.7 ± 9.3	11.2 ± 6.1	0.778
30 days mortality, %	9 (1.9)	1 (5.0)	0.873
90 days mortality, %	11 (2.3)	1 (5.0)	0.979

Abbreviations: LN = lymph node, LOS = length of stay. Categorical data are expressed as number (%) and continuous data as mean ± SD or median (interquartile range).

**Table 4 cancers-14-05683-t004:** Clinical characteristics and perioperative outcomes between Open and VATS pneumonectomy for pT 2 tumors. and pT 3-4 tumors.

Characteristic	pT 2			pT 3-4		
	Open n = 248	VATS n = 53	*p*	Open n = 230	VATS n = 32	*p*
Operative time, min	142.2 ± 49.8	191.0 ± 75.2	<0.001	137.8 ± 41.8	200.5 ± 62.1	<0.001
Blood loss, ml	200.6 ± 182.4	248.1 ± 274.7	0.233	241.7 ± 319.2	185.9 ± 99.4	0.328
Number of dissected LN	16.9 ± 7.4	17.0 ± 8.6	0.828	17.1 ± 6.8	20.0 ± 7.7	0.054
R0 Surgery, %	229 (92.3)	49 (92.5)	1.000	210 (91.3)	30 (93.8)	0.899
Major complications, %	51 (20.6)	4 (7.6)	0.042	35 (15.2)	6 (18.7)	0.606
Atrial Fibrillation	26 (10.5)	2 (3.8)		16 (7.0)	1 (3.1)	
Bleeding	8 (3.2)	0		3 (1.3)	1 (3.1)	
ARDS	7 (2.8)	0		3 (1.3)	0	
Infection	4 (1.6)	2 (3.8)		5 (2.2)	0	
Bronchopleural Fistula	4 (1.6)	0	1.000	4 (1.7)	4 (12.5)	0.006
Chylothorax	2 (0.8)	0		1 (0.4)	0	
Esophageal injury	0	0		1 (0.4)	0	
Gastrointestinal bleeding	0	0		2 (0.9)	0	
Days in ICU	3.3 ± 5.2	2.4 ± 1.9	0.190	2.6 ± 2.1	3.8 ± 7.4	0.052
LOS, day	12.4 ± 11.6	10.2 ± 4.9	0.189	11.1 ± 5.7	10.3 ± 6.8	0.478
30 days mortality, %	6 (2.4)	0	0.595	3 (1.3)	2 (46.3)	0.220
90 days mortality, %	8 (3.2)	0	0.359	3 (1.3)	3 (9.4)	0.026

Abbreviations: LN = lymph node, LOS = length of stay. Categorical data are expressed as number (%) and continuous data as mean ± SD or median (interquartile range).

**Table 5 cancers-14-05683-t005:** Clinical characteristics and perioperative outcomes between Open and VATS pneumonectomy on the right and the left side.

Characteristic	Right			Left		
	Open n = 77	VATS n = 24	*p*	Open n = 401	VATS n = 61	*p*
pT Stage			0.456			0.069
2	35 (45.5)	13 (54.2)		213 (53.1)	40 (65.6)	
3–4	42 (54.5)	11 (45.8)		188 (46.9)	21 (34.4)	
Operative time, min	156.5 ± 46.0	209.6 ± 59.7	0.000	136.9 ± 45.5	188.6 ± 73.7	0.000
Blood loss, ml	261.0 ± 350.2	177.1 ± 95.5	0.250	212.6 ± 236.0	243.4 ± 258.9	0.093
Number of dissected LN	18.0 ± 7.7	17.9 ± 5.5	0.943	16.8 ± 7.0	18.2 ± 9.2	0.241
R0 Surgery, %	63 (81.8)	22 (91.7)	0.404	376 (93.8)	57 (93.4)	0.923
Major complications, %	19 (24.7)	5 (20.8)	0.699	67 (16.7)	5 (8.2)	0.088
Atrial Fibrillation	4 (5.2)	1 (3.6)		38 (9.5)	2 (5.2)	
Bleeding	5 (6.5)	0		6 (1.5)	1 (1)	
ARDS	1 (1.3)	0		9 (2.2)	0	
Infection	2 (2.6)	0		7 (1.7)	2 (3.9)	
Bronchopleural Fistula	5 (6.5)	4 (16.7)	0.127	3 (0.7)	0	1.000
Chylothorax	1 (1.3)	0		2 (0.5)	0	
Esophageal injury	0	0		1 (0.2)	0	
Gastrointestinal bleeding	1 (1.3)	0		1 (0.2)	0	
Days in ICU	2.9 ± 2.6	2.6 ± 8.4	0.333	3.0 ± 4.3	2.2 ± 1.9	0.168
LOS, day	12.4 ± 8.4	10.8 ± 7.4	0.396	11.6 ± 9.4	10.0 ± 4.8	0.203
30 days mortality, %	3 (3.9)	1 (4.2)	1.000	6 (1.5)	1 (1.6)	1.000
90 days mortality, %	3 (3.9)	2 (8.3)	0.737	8 (2.0)	1 (1.6)	1.000

Abbreviations: LN=lymph node, LOS=length of stay. Categorical data are expressed as number (%) and continuous data as mean ± SD or median (interquartile range).

## Data Availability

The data presented in this study are available on request from the corresponding author.

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
