# Peer review of "Surgical Outcomes of Video-Assisted versus Open Pneumonectomy for Lung Cancer: A Real-World Study"

_cancers, 2022, doi:10.3390/cancers14225683_

Round 1

Reviewer 1 Report

We salute you for your valuable study.

Thoracoscopic pneumonectomy is not versatile and has little merit as a minimally invasive procedure. However, this paper is based on the experience of a single institution with a very large number of total pneumonectomies, and I believe that it is worthwhile to publish the results of this study.

It is valuable for respiratory surgeons to be shown the advantages and disadvantages of this approach.

I think it would be good to revise the paper by adding a description of the Conversion and other information that would be clinically important.

1) Please provide a breakdown of Conversion reasons. We believe this would be very informative for clinicians.

2) Although OS is shown in Figure 2, cancer specific survival should also be shown. This information is necessary from the viewpoint of whether minimally invasive surgery has provided adequate cancer control.

3) Many patients with stage I lung cancer are included, why did these patients need pneumonectomy? 

4) What is the `tumor location` used for matching? Is it peripheral or central localization? Or left and right?

5) Although not statistically significant, the frequency of bronchopleural fistulas after thoracoscopic right total pneumonectomy was 16%, more than double the presence of thoracotomy. Since bronchopleural fistula is a lethal complication, this difference should not be ignored. Please discuss this point.

Reviewer 2 Report

The manuscript by Luo and colleagues represents a retrospective study of video-assisted versus open pneumonectomy for lung cancer. The authors performed a retrospective, mono centric study of patients underwent VATS and open pneumonectomy in Shanghai Chest Hospital between1990 and 2020. The authors showed a potential benefit of VATS for patient with early stage lung cancer requiring pneumonectomy regarding the postoperative complications. They showed that operation with VATS needed more time compared to open pneumonectomy.

I have the following concerns:

-        What were the selection criteria for the patients in the VATS groups

-         I suggest using a classification system for postoperative complication like the Clavien-Dindo classification.

-        Please describe the technique for VATS pneumonectomy: the incision, how many ports?, how you got the removed lung outside the thorax, and the pathology report.  

-        What was the oncological outcome regarding recurrence and overall survival? Could you justify the VATS pneumonectomy when it takes longer time with a good risk for conversion into open pneumonectomy without benefit regarding the recorded postoperative complications in the whole cohort.

Reviewer 3 Report

The authors have conducted a study to compare video-assisted surgery (VATS) versus open pneumonectomy for lung cancer, showing the benefits of VATS. This is an interesting work as it shows that the complication rate in the VATS group was significantly lower than that in the in pT1-2 stage tumors. 

However, it is necessary to make minor corrections:

Lane 114: p must be lowercase and may be italicized.

Table 1: check, the table overlaps with the lane 152.Add space between mean and SD.Add space between “,” and “%”.

Tables 2, 3, 4 and 5: Add space between mean and SD.

Figure 2: To separate () in the title axes.

Reviewer 4 Report

This retrospective analysis focused on the outcome difference of VATS or open pneumonectomy in patients with lung cancer.

1. There was no explanation of the definition of VATS and the open approach for pneumonectomy.

2. Please add information on reasons for pneumonectomy. Pneumonectomy for T1 lung cancer is difficult to understand.

3. How did the authors calculate the number of dissected LNs? Did the authors count two lymph nodes if the operator divided one LN into two LNs by procedure? The number of per station basis should also be mentioned.

4. It is unclear that the mean operative time was 190.2 min for all VATS cases, but both VATS operative times for pT1-2 and pT3-4 were more than 190.2 min (191.0 and 200.5 min retrospectively).

5. To know the indication of pneumonectomy, N factor information will be mandatory.

6. From the previous reports, preserving lung parenchyma as much as possible by plastic procedure showed a better outcome than pneumonectomy. Were there any cases in which the potential to preserve lung parenchyma by open approach with the plastic procedures?

Round 2

Reviewer 1 Report

Appropriate revisions have been made to this paper. I judge that the content is worthy of publication

Reviewer 2 Report

thanks for your response.

Reviewer 4 Report

The queries were addressed.